# A Study on the Evaluation of Flow Distribution Evenness in Parallel-Arrayed-Type Low-Pressure Membrane Module Piping

**DOI:** 10.3390/membranes11100751

**Published:** 2021-09-30

**Authors:** No-Suk Park, Sukmin Yoon, Woochang Jeong, Yong-Wook Jeong

**Affiliations:** 1Department of Civil Engineering and Engineering Research Institute, Gyeongsang National University, 501, Jinju-Daero, Jinju 52828, Korea; nspark@gnu.ac.kr (N.-S.P.); gnuysum@gmail.com (S.Y.); 2Department of Civil Engineering, Kyungnam University, 7, Kyungnamdaehak-ro, Masanhappo-gu, Changwon 51767, Korea; jeongwc@kyungnam.ac.kr; 3Department of Architecture, Sejong University, 209, Neungdong-ro, Gwangjin-gu, Seoul 05006, Korea

**Keywords:** flow distribution evenness, manifold pipe structure, ultrasonic flow meter, computational fluid dynamics, Reynolds number

## Abstract

The objectives of this study were to measure the flow rate distribution from a header pipe to each module installed in parallel for a water treatment membrane filtration process in operation and to investigate the reason for an uneven distribution of the flow rate via the CFD technique. In addition, this study attempted to propose the ratio of the branch pipe to the header pipe required to equalize the flow distribution for the same membrane filtration process. Finally, the relationship between the Reynolds number in the header pipe and the degree of the manifold flow distribution evenness was investigated. Mobile ultrasonic flow meter was used to measure the flow rate flowing from the membrane module pipe to each module, and the CFD technique was used to verify this. From the results of the actual measurement using ultrasonic flow meter and CFD simulation, it was confirmed that the outflow flow rate from the branch pipe located at the end of the header pipe was three times higher than that of the branch pipe near the inlet. The reason was that the differential pressure generated between each membrane module was higher toward the end of the header pipe. When the ratio of the sum of the cross-sectional area of the branch pipe and the cross-sectional area of the header pipe was reduced by about 30 times, it was confirmed that the flow rate flowing from each branch pipe to the membrane module was almost equal. Also, if the flow in the header pipe is transitional or laminar (Reynolds No. is approximately 4000 or less), the flow rate flowing from each branch pipe to the membrane module can be more even.

## 1. Introduction

Parallel-arrayed-type low-pressure membrane processes, such as microfiltration and ultrafiltration, have recently been increasingly applied in drinking water treatment systems. These low-pressure membrane processes selectively remove particulate contaminants larger than membrane pores by a straining mechanism using the differential pressure between the pores [1,2]. Membrane filtration is a widely used technology because it can remove particulate matter, organic matter, inorganic salt, etc., and can produce stable water quality, depending on the size of the pores [3,4]. However, despite the many advantages of membrane filtration technology, the reduction in filtration efficiency over operating time, that is, the periodic occurrence of membrane fouling, has been identified as a major problem in the introduction of membrane filtration technology into water treatment fields [3,4,5,6]. In general, raw water flows into each membrane module installed parallel to the header pipe. The filtered water treated by the membrane gathers at the top and flows out (refer to Figure 1). It has a structure, such that the water to be treated flows from a relatively large header pipe at the bottom to a membrane module installed in parallel through upward manifold pipes [1,2]. 

This structure has been widely applied in thermodynamic cooling. A method for uniformly introducing a refrigerant into cooling systems was proposed by Shen et al. and Datta and Majumdar in the early 1990s [7,8,9]. They experimentally deduced that the smaller the ratio of the cross-sectional areas of the manifold branch pipes to that of the header pipe, the more even the flow rate distribution. In addition, Eguchi et al. found the advantage that the smaller the ratio of the cross-sectional area of the manifold to the header pipe, the smaller the loss factor (the ratio of the average power loss to the peak load loss) [10]. 

However, it is difficult to adopt this method in membrane processes applied to water treatment. Because the inlet diameter (diameter of the manifold branch pipe) of commercial low-pressure membrane modules is typically 50 mm or more, a limitation exists in reducing the ratio of the header pipe to the cross-sectional area of the distribution manifold to be connected. When the ratio is small, a problem emerges, such that the head loss and energy consumption increase. This study presents the result obtained from reviewing this, as well as the ratio of the branch pipe to the header pipe required to equalize the flow rate for the membrane filtration process in operation.

Via numerical analysis, Hong and Riggs emphasized that a tapered header pipe with a gradually reduced cross-sectional area obtains a more uniform flow velocity distribution than a distribution pipe with a constant cross-section; hence, the hydraulic pressure becomes constant [11]. However, this method is applicable when the flow velocity of the header pipe is relatively low. The header piping applied to the actual membrane filtration process for drinking water treatment is generally designed and operated to maintain a flow rate of 1.0 m/s or more. Muhana and Novog investigated the effect of the header pipe flow rate and Reynolds number on the flow distribution through each manifold branch pipe for a header pipe. They experimentally determined that as the Reynolds number increased, the flow of the manifold branch pipe on the far side from the header pipe inlet increased [12]. Considering that the header pipe is operated by pipe flow, this study attempted to elucidate the relationship between the Reynolds number in the header pipe and the degree of evenness of the manifold flow distribution.

Since the late 2000s, studies have been conducted to optimize the designs in the field where the manifold is applied via the computational fluid dynamics (CFD) technique. Via CFD and verification experiments for modular membrane systems, Ding investigated the occurrence of significant unevenness in the flow rate flowing from the header pipe to each membrane module [13]. In addition, via CFD simulations, Paul et al. inferred that the flow rate of fluid flowing into the parallel-arrayed unit cells in the stack of proton exchange membrane fuel cells is related to the flow direction of the outflow header [14]. As described above, although the studies on the flow distribution of header pipes with manifolds have significantly progressed since the late 1990s, most of them are achievements in fuel cell and energy engineering fields. There has been almost no research on membrane module piping in the water treatment field. 

Therefore, in this study, the flow rate distribution from the header pipe to each module installed in parallel was measured with a mobile ultrasonic flow meter for a water treatment membrane filtration process in operation, and the reason for the uneven distribution of the flow rate was investigated via the CFD technique. In addition, this study attempted to propose the ratio of the branch pipe to the header pipe required to equalize the flow distribution for the same membrane filtration process. Finally, the relationship between the Reynolds number in the header pipe and the degree of evenness of the manifold flow distribution was investigated.

## 2. Research Methods

### 2.1. Membrane Filtration Process for Drinking Water Treatment

The membrane filtration facility selected for this study is located within the G_treatment plant in the Republic of Korea, with a maximum capacity of 30,000 m^3^/day. The specifications of the membrane are shown in Table 1, and the membrane module is a microfiltration membrane. The membrane filtration flow rate (Flux) was designed to be 1.0 m^3^/m^2^·day under normal conditions, and when one series is stopped for backwashing, the filtration flow rate of the remaining three series is increased to 1.33 times during the normal operation to retain a constant flow rate. The membrane filtration system is equipped with 4 series, 24 units (6 units per series), and 480 modules (20 membrane modules per unit), and each unit has a total of 20 modules arranged symmetrically (refer to Figure 1 and Figure 2). The membrane filtration process is operated with 50 s of water inlet, 30 min of membrane filtration, 30 s of backwashing (air + water), and 45 s of drainage, and CEB (Chemical-Enhanced Backwashing) and CIP (Clean-in-Place Instructions) are periodically conducted.

Figure 2 solely illustrates the lower part of the inlet header pipe, as presented in Figure 1. Ten membrane modules are installed at an interval of 0.40 m at a position of 0.35 m from the inlet, and the distance between the last tenth of the pipe and the end of the header pipe is 0.35 m, identical with the inlet part. The average daily operating flow for one inlet header pipe unit is approximately 38.96 m^3^/hr.

### 2.2. Flow Distribution Measurement Using an Ultrasonic Flow Meter

In this study, the flow rate into each module was measured using a clamp-on-type (dry-type method) ultrasonic flow meter. An ultrasonic flow meter measures the flow rate in a pipeline using the characteristics of the ultrasonic waves. Recently, its application field has expanded because the clamp-on type (dry-type method) is easier to install than the wet type method, and it is easy to manage because there is no damage to the pipe. The wet-type method has a relatively better accuracy than the dry-type method. However, the dry-type method is preferred because the wet-type has poor installation and mobility issues. Table 2 summarizes the specifications of the transducer and flow meter used in this study.

To accurately measure the flow rate, a distance of 5D or more from a curved pipe is required [15]. However, the distance between the branch pipes is only about 0.40 m in the case of the selected membrane module inlet header pipe. Therefore, it is difficult to obtain accurate flow rate data. As illustrated in Figure 3a, a method was adopted by which the flow rate data were read directly from the header pipe, and the amount of change in flow that occurred as it passed through each branch pipe was considered the inflow flow rate of the branch pipe. As mentioned in the measurement results, the flow rate data were relatively unstable; however, stable data were obtained approximately 5 min after installation.

### 2.3. Statistical Processing of Experimental Data

The G_water treatment plant comprises six units, and the average daily operating flow rate for each header pipe (one unit) is calculated as 55 m^3^/day. However, it is difficult to directly compare and analyze the flow distribution of manifold pipes by units, because the inflow rate for each unit is not constant due to changes in the operating conditions. Therefore, minimum–maximum normalization was adopted to quantitatively evaluate the flow distribution of the manifold pipe for each unit obtained by the ultrasonic flow meter. The maximum and minimum values for the flow rates flowing into each branch pipe from one unit process were derived, and the flow rate was normalized using the procedure of Equation (1):(1)$$z_{i}=\frac{x_{i}-x_{min}}{x_{max}-x_{min}}$$
where $x_{i}$ is flow rate data, $x_{min}$ and $x_{max}$ are minimum and maximum values, respectively, and $z_{i}$ is the normalized values.

To analyze the distribution of the normalized branch pipe flow in each unit, linear regression analysis was performed using the dimensionless distance of the branch pipe position and normalized flow data. The dimensionless distance of the branch pipe represents the relative distance measured to each branch pipe from the position of the first branch pipe as the origin. In addition, dimensionless distance as an explanatory variable and normalized flow values as a response variable were inputted in order to analyze the flow distribution evenness of each manifold pipe.

### 2.4. Methodology of CFD Simulations

In this study, the commercial CFD software ANSYS CFX 16.0 was used to simulate the evenness of the water distribution within the header and manifold pipes [16,17]. Figure 4 presents the shape of the header and manifold pipes for simulating the flow behavior. Figure 4a,b indicates that the ratio of the sum of the cross-sectional areas of the branch pipe and header pipe changes from 3.265 to 0.116 by increasing the diameter of the header pipe from 0.14 m to 0.74 m. If this ratio is reduced, it is assumed that the equalization of the flow rate will be improved. This is the area ratio determined by the trial and error method while increasing the diameter of the header pipe gradually. 

The length of the pipe is 9.17 m in both cases (a) and (b), and the diameter and distance between the branch pipes are identical. The diameter of the branch pipe is 80 mm, and the direction of the branch pipe outlet is upward. The CFD simulations were conducted, assuming a steady state, and the treated fluid was assumed to be 25 °C water, considering the room temperature condition. 

The CFD simulation was performed by splitting the geometry of interest into numerous elements, collectively known as grids or cells. Subsequently, the momentum and continuity equations were formulated for each grid together with the given boundary conditions, and they were repeatedly solved by using the finite volume method.

The time-averaged Navier–Stokes equations for momentum and continuity were solved in this study for achieving a steady, incompressible, turbulent, and isothermal flow. The continuity and momentum equations are, respectively, [17,18]:(2)$$\underline{\nabla }\cdot \left(\underline{U}\right)=0$$
(3)$$\underline{\nabla }\cdot \left(\rho \underline{U}⊗\underline{U}-\mu \underline{\nabla U}\right)=\underline{B}+\underline{\nabla }P-\underline{\nabla }\cdot (\bar{\rho \underline{u}⊗\underline{u}})$$
where *ρ* and *μ* are the fluid density and dynamic viscosity, respectively, *P* is the pressure, *U* is the fluid mean velocity, *B* is the body force, and *u* is the fluctuating velocity. 

The authors assumed that the turbulence in the pipes is isotropic. Therefore, a standard k–ε model was used for modelling the turbulence transport of the momentum. At the pipe wall surface, a no-slip condition was assumed, and a widely used standard wall boundary method was applied to bridge the viscous sublayer. Therefore, it was assumed that the velocity of the component at each wall is zero. The wall shear stress was obtained from the logarithmic law of the wall [1].

## 3. Results and Discussion

### 3.1. Flow Distribution Measurement Results

Figure 5 presents the actual measurement of the flow distribution in the four series of the G_membrane filtration plant. The x-axis represents the length of the header pipe as the distance from the first inlet. The total length of the header pipe (4.3 m) was expressed by subtracting 0.35 m from the inlet to the first branch pipe. The y-axis represents the data measured by an ultrasonic flow meter. These include the flow rate flowing into each membrane module from the header pipe. All membrane modules in Unit 1 and 3 were replaced in 2020, and those in Unit 2 were replaced in 2019; however, the modules in Unit 4 have been operating continuously since their installation in 2010. It was observed that the flow rate flowing into the membrane module through the branch pipe increased as the distance from the inlet increased. Although the ultrasonic flow meter could not accurately measure the flow values, the trend was sufficiently readable. During the membrane filtration process, frequent flow rate fluctuations and backwashing were performed periodically; hence, there was a limit to obtaining accurate data. Nevertheless, it could be observed that the flow rate increased as the branch pipe was located closer to the end of the header pipe, instead of being closer to the inlet. The flow rates from the first branch pipe in the inlet and the branch pipe located at the end of the header pipe differed approximately by a factor of 3. 

Figure 6 presents the results of min–max normalization and linear regression analysis using data acquired with an ultrasonic flow meter, as mentioned in Section 2.3. The x- and y-axis data are expressed as dimensionless distances and dimensionless flow rates. Linear regression equations were expressed using linear and tertiary equations, respectively. Although the coefficient of determination (R^2^) is relatively low, it exhibits a positive slope. Hence, the upward trend of the manifold pipe flow rate can be clearly observed. In the regression analysis results using a cubic polynomial, it can be inferred that two inflection points occur at dimensionless distances of 0.3 and 0.75, respectively. These inflection points increase at the dimensionless distance of 0~0.3, decrease slightly at 0.3~0.75, and then rapidly increase thereafter.

### 3.2. Results of CFD Simulations

Figure 7 presents the simulation results obtained via CFD technique when the inflow flow rate was 38.96 m^3^/hr for the inlet manifold pipe structure (Figure 4a) of the membrane filtration process in operation. Here, the inlet is to the right, and the main flow of raw water flows from right to left.

Figure 8 presents the flow rate from each branch pipe obtained via the CFD simulation. The flow rate measured using an ultrasonic flow meter was similar to the flow rate obtained from the CFD simulation. Although it exhibits a difference of approximately 10% from the measured value, the flow rate flowing out through the branch pipe increases toward the end compared to that near the inlet of the header pipe. In addition, similar to the actual measurement, it was verified that the flow rates of the outflow from the first branch pipe in the inlet and the branch pipe located at the end of the header pipe differed approximately by a factor of 4. In particular, the instantaneous decrease in the flow rate at the 0.3 and 0.75 positions on the dimensionless distance was almost the same as the position showing the inflection point in the linear regression analysis graph with a third-order polynomial. However, the authors could not identify the reason for the relatively larger flow rate from the branch pipe near the end of the header pipe with such velocity and flow distribution patterns.

As illustrated in Figure 9, the pressure distribution at the end of the header pipe was closely examined. The figure presents an enlarged view of the pressure distribution around the nine branch pipes at the right end of the header pipe. “Pressure” represents the distribution of the total pressure, including the dynamic and static pressures. Because the header pipe has a diameter of 0.14 m and is horizontally arranged, the static pressure can be almost negligible. As illustrated in the Figure 9, it can be observed that the pressure increases relatively toward the end of the right header pipe. Assuming that the permeability coefficients (K) representing the resistance of each membrane module are almost the same, this can be attributed to “Darcy’s law”. Considering the pressure difference (head loss) between the pressure of the inlet and outlet of the membrane module, the head loss is approximately twice as large in the branch pipe located at the end of the header pipe compared to the branch pipe at the inlet side of the header pipe. Based on Darcy’s law, the velocity (V) through the membrane module increases; hence, the flow rate increases toward the end of the header pipe, according to the continuity Equation (4):
(4)$$V=\frac{Q}{A}=-K\;\frac{\mathrm{dh}}{\mathrm{dL}}$$
where V and Q are flow velocity and flow rate, respectively, A is the cross-sectional area of the membrane module, K is the permeability coefficient of the membrane module, dh is the pressure difference (differential pressure) between the inlet and outlet of each membrane module, and dL is the length of the membrane module.

The pressure difference generated by the membrane module toward the end of the header pipe explains the increase in the branch pipe flow rate toward the end of the header pipe, instead of near the inlet in the parallel-arrayed membrane modules. The method of achieving even flow distribution by applying a tapered header pipe with a reduced cross-sectional area, which has been proposed in the energy engineering field [11], cannot be applied to the actual water treatment membrane module piping structure. 

### 3.3. Header Pipe Cross-Sectional Area Expansion 

As mentioned above, an additional objective of this study was to determine the ratio of the branch pipe to the header pipe required to equalize the flow distribution for the membrane filtration process. Furthermore, the authors attempted to clarify the relationship between the Reynolds number in the header pipe and the degree of flow distribution evenness in the manifold pipe. According to the flow rate (flow velocity) into the header pipe via actual measurements, the flow rate of the branch pipe close to the inlet can be categorized depending on whether the flow rate of the branch pipe at the end of the pipe is relatively large or small (see Figure 10).

The flow rate from the branch pipe at the end (right side in Figure 10) was high because the inlet flow velocity in the header pipe of the actual membrane filtration facility of the G_water treatment plant was relatively high. According to the continuity equation, if the diameter or cross-sectional area of the header pipe increases, the flow velocity in the pipe could decrease. Accordingly, CFD simulations were performed while increasing the diameter of the header pipe gradually, under the assumption that the flow rate from each branch pipe could be found to be even. The obtained results were derived, as presented in Figure 11. When the ratio of the sum of the cross-sectional area of the branch pipe and header pipe decreases by approximately 30 times ($\frac{\sum a}{A}=0.116$) compared to the case ($\frac{\sum a}{A}=3.265$) of the actual manifold pipe, the flow rate into each membrane module is almost even. 

In addition, the Reynolds number can be obtained by dividing the inflow flow rate of 38.96 m^3^/hr into the header pipe by the cross-sectional area of the pipe, which are calculated as 98,420 and 262 in Figure 11a,b, respectively. If the inlet velocity and Reynolds number are decreased by increasing the diameter of the header pipe more than this, as is the case in Figure 11b, the flow distribution pattern is reversed, as a relatively high flow out from the branch pipe close to the inlet appears in the opposite direction of the flow distribution. In Figure 11b, the reason the total pressure is the same is that the hydraulic pressure becomes equal as the diameter of the header pipe increases.

Generally, in pipe flow, if the Reynolds number is over 4000, it is classified as turbulent flow; if it is less than 2400, it is classified as laminar; and the flow with a value between these is classified as transition flow. In this study, although a conclusion was reached for an actual membrane filtration facility, if the flow on the header pipe is transitional or laminar, the flow distribution into the membrane module can be equalized. However, as illustrated in Figure 11b, if the mechanical installation and space problems are considered, limitations will emerge against the method of increasing the size of the header pipe to improve flow distribution evenness. In addition, the initial investment and energy costs for the operation are also expected to increase.

## 4. Conclusions

In this study, the flow distribution in each module was measured for the filtration process of the parallel-arrayed low-pressure membrane (microfiltration) during the operation of actual water treatment. In addition, the authors attempted to determine the cause via the computational fluid dynamics (CFD) technique and suggest the ratio of branch and header pipes required to improve the flow distribution evenness. Finally, the relationship between the Reynolds number in the header pipe and the degree of evenness of the manifold flow distribution was investigated. The obtained results are summarized as follows:(1)From the actual measurement using an ultrasonic flow meter and the obtained CFD simulation results, it was verified that the flow rate increased toward the end of the header pipe, instead of the branch pipe close to the inlet in the header pipe of the membrane units. The flow rate from the first branch pipe in the inlet and the branch pipe located at the end of the header pipe differed approximately by a factor of 3.(2)The outflow into membrane modules increased toward the end of the header pipe because the pressure difference between each membrane module increased toward the end of the header pipe.

When the ratio of the sum of the cross-sectional areas of the branch and the header pipes was reduced by approximately 30 times, the flow distribution into 10 membrane modules from each branch pipe was almost even. Therefore, considering the Reynolds number in the header pipe, the flow distribution into the membrane module can be equalized when the flow in the pipe is transitional or laminar (Reynolds number is approximately 4000 or less).

## Figures and Tables

**Figure 1 membranes-11-00751-f001:**
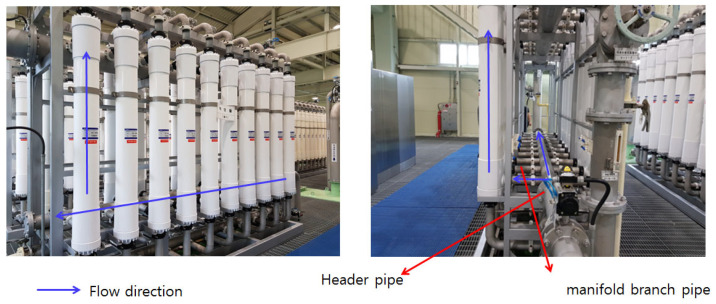
Actual parallel-arrayed low-pressure membrane module for drinking water treatment.

**Figure 2 membranes-11-00751-f002:**
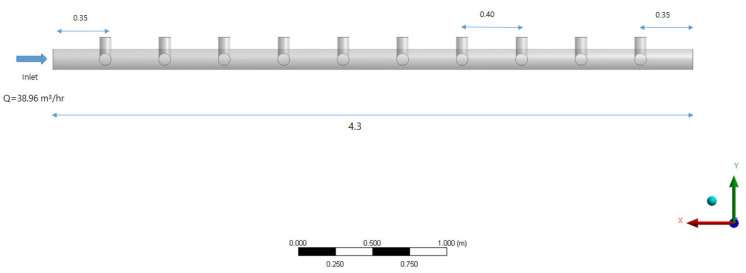
Geometry of inlet header and manifold pipes (unit: m).

**Figure 3 membranes-11-00751-f003:**
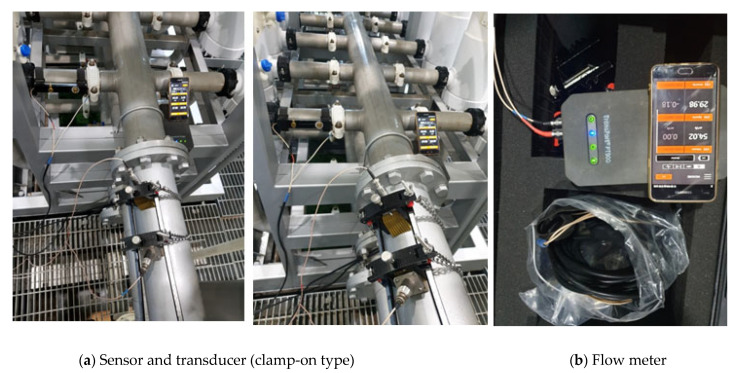
Clamp-on transducer and ultrasonic liquid flow meter adopted in this research. (**a**) Sensor and transducer (clamp-on type); (**b**) Flow meter.

**Figure 4 membranes-11-00751-f004:**
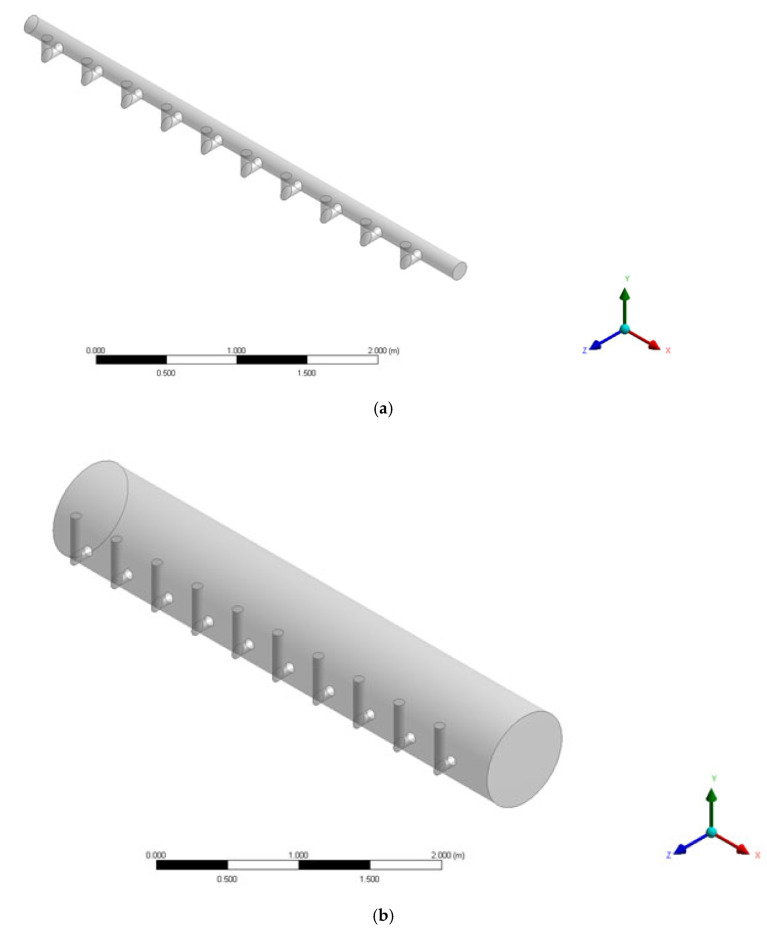
Geometry of manifold pipes: (**a**) Actual manifold pipe (Σa/A=3.265, a: branch pipe cross-sectional area, A: header pipe cross-sectional area) and (**b**) Manifold- pipe with an enlarged diameter (Σa/A = 0.116).

**Figure 5 membranes-11-00751-f005:**
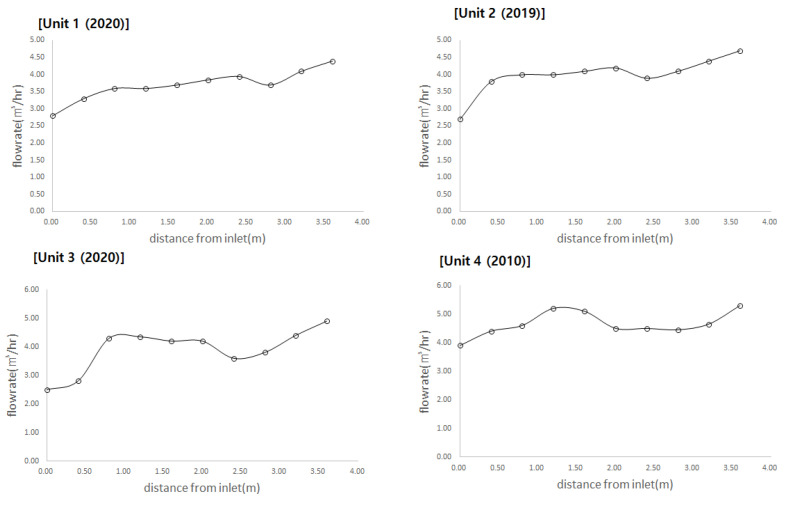
Flow distribution measurement results.

**Figure 6 membranes-11-00751-f006:**
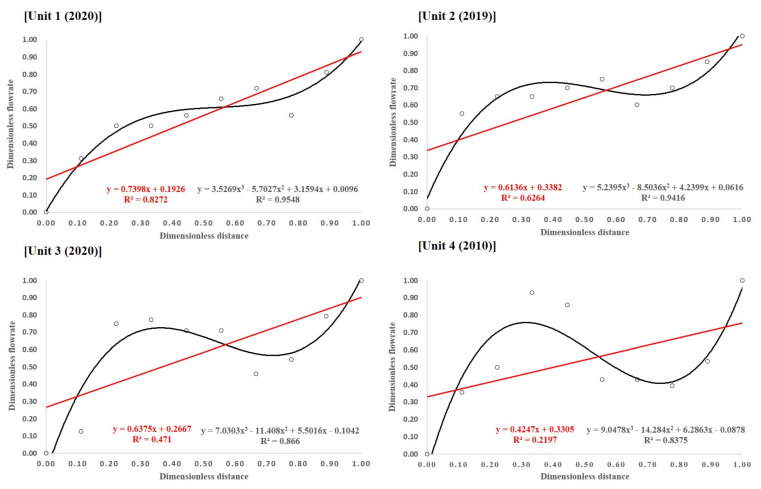
Statistical analysis results.

**Figure 7 membranes-11-00751-f007:**
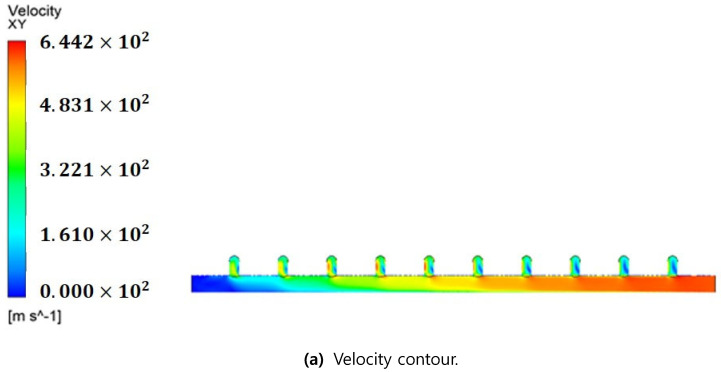
CFD simulation results for actual manifold pipe. (**a**) Velocity contour; (**b**) Velocity vector.

**Figure 8 membranes-11-00751-f008:**
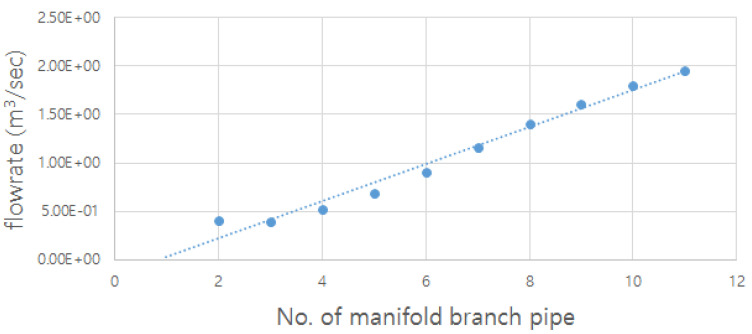
Flow distribution pattern from CFD simulation results.

**Figure 9 membranes-11-00751-f009:**
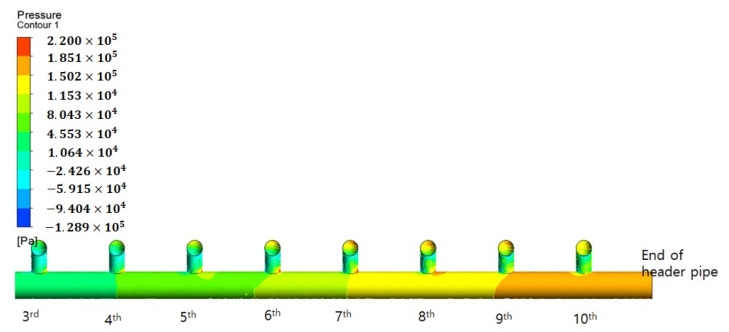
Pressure distribution.

**Figure 10 membranes-11-00751-f010:**
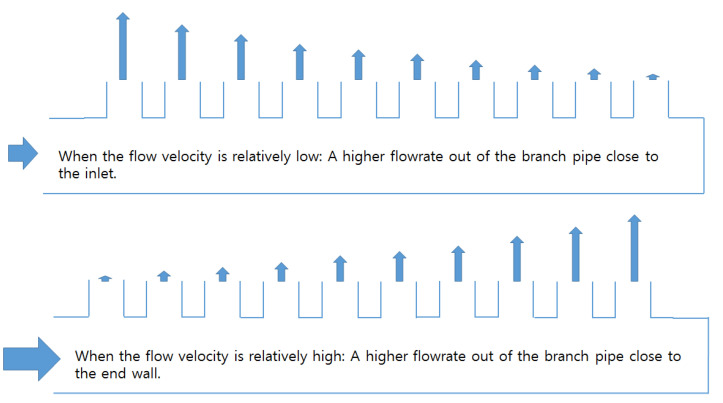
Relationship between header pipe inlet velocity and flow distribution pattern.

**Figure 11 membranes-11-00751-f011:**
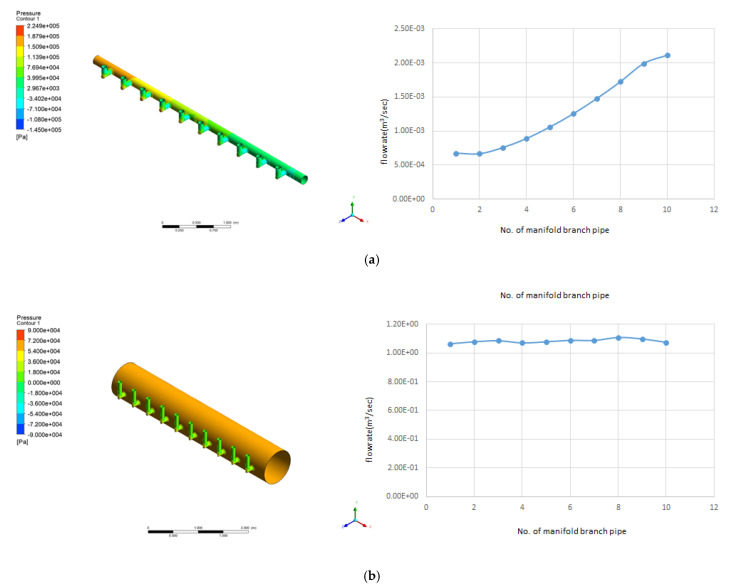
Improving flow distribution evenness via the expansion of the header pipe’s cross-sectional area: (**a**) CFD simulation results for actual manifold pipe ($\frac{\sum a}{A}=3.265,\;a$: cross-sectional area of branch pipe, A: cross-sectional area of header pipe) and (**b**) CFD simulation results of manifold pipe with enlarged diameter ($\frac{\sum a}{A}=0.116$).

**Table 1 membranes-11-00751-t001:** Membrane module specifications.

Membrane manufacturer	Toray, HFS-2020
Membrane type	Microfiltration (MF)
Membrane module shape	External-pressure-type hollow fiber membrane (casing)
Hollow fiber	Inner D 0.9 mm/external D 1.4 mm
Pore size	0.05 µm
Membrane material	PVDF
Flux	(Ordinary) 1.0 m^3^/m^2^·day (Max.) 1.33 m^3^/m^2^·day
Module Size	D 216 mm × L 2160 mm
Membrane area	72 m^2^/module
Allowable pressure	300 kPa
Allowable pH	1~10 at filtration, 1~12 at chemical cleaning

**Table 2 membranes-11-00751-t002:** Ultrasonic flow meter and transducer specifications.

Flow Meter	PT878	Transducer Type	Clamp-On
Flow type	All acoustically conductive fluids	Applications	Liquid
Pipe size	12.7 mm~7.6 m	Compatible meters	PT878
Pipe wall thickness	Up to 76.2 mm	Frequency	1MHz
Pipe materials	All metals and most plastics	Process temp.	−20–210 °C
Repeatability	±0.1% to 0.3% of reading	Ambient temp.	−20–40 °C
Range	−12.2 to 12.2 m/s	Materials of construction	Metals and plastics
Range ability	400:1	-	-
Measurement parameters	Volumetric flow, totalized flow, and flow velocity	-	-

## Data Availability

Not applicable.

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
