# Peer review of "A Study on the Evaluation of Flow Distribution Evenness in Parallel-Arrayed-Type Low-Pressure Membrane Module Piping"

_membranes, 2021, doi:10.3390/membranes11100751_

Round 1

Reviewer 1 Report

  • What problem was studied and why is it impact? What is the novelty of the work and where does it go to beyond previous effects in the literature.

  • Revise the abstract section to accurately present not only the aim of the study and methodology. Revise the abstract to present (a) significance of the study, (b) aim of the research study, (c) research methodology, and (d) major conclusion drawn from the study.

  • clarify and discuss the motivation of this study, and the novelty and the significance of the results obtained here and compare them with those available in the literature, also including discussions on potential applications.
  • Authors must include the values of all physical parameters considered.

  • Some information is missing in the references section, and hence authors should need to modify it.

  • What software is used for the simulations? Was the code for the method implemented by the authors or a function already existing in the software was used? If the code for the numerical method was taken from another publication or is part of the software used, please cite the resource.
  • You should add appropriate references for governing equation.
  • Need more discussion about the results? In the results section, the author needs to analyse the finding by giving reasons for each fact. Please explain every point?
  •  

Author Response

Authors’ Reply to Reviewers Comments

Manuscript ID: membranes-1394488

Title: A Study on the Evaluation of Flow Distribution Evenness in Parallel-arrayed type Low Pressure Membrane Module Piping

Authors: No-Suk Park, Sukmin Yoon, Woochang Jeong, Yong-Wook Jeong *

The authors acknowledge the valuable comments and suggestions provided by the editor and reviewers. The authors wish to inform the editor and reviewers that we have revised and improved the original manuscript taking into account all of the concerns and comments expressed by the editor and reviewers. Modifications that have been made in the revised manuscript are all highlighted in yellow print.

Reviewer 1

Comment 1:

  1. What problem was studied and why is it impact? What is the novelty of the work and where does it go to beyond previous effects in the literature.

Reply:

- The authors revealed that the inflow from the header pipe to each module installed in parallel was uneven in the water treatment membrane filtration process.

- Flow unevenness causes a serious bias in membrane module fouling and inefficient backwashing.

- It is not the purpose of this paper to propose a new innovative method for flow equalization. The main purpose is to elucidate the cause of flow unevenness in detail.    

Comment 2:

  1. Revise the abstract section to accurately present not only the aim of the study and methodology. Revise the abstract to present (a) significance of the study, (b) aim of the research study, (c) research methodology, and (d) major conclusion drawn from the study.

Reply:

- The authors revised as your comments,

(Objectives) : The objectives of this study were to measure the flow rate distribution from the header pipe to each module installed in parallel for the water treatment membrane filtration process in operation, and to investigate the reason for the uneven distribution of the flow rate via the CFD technique. In addition, this study attempted to propose the ratio of the branch pipe to the header pipe required to equalize the flow distribution for the same membrane filtration process. Finally, the relationship between the Reynolds number in the header pipe and the degree of the manifold flow distribution evenness was investigated.

(Methods): Mobile ultrasonic flow meter was used to measure the flow rate flowing from the membrane module pipe to each module, and the CFD technique was used to verify this.

(Results and Discussion): From the results of the actual measurement using ultrasonic flowmeter and CFD simulation, it was confirmed that the outflow flow rate from the branch pipe located at the end of the header pipe was three times higher than that of the branch pipe near the inlet. The reason was that the differential pressure generated between each membrane module was higher toward the end of the header pipe.

(Conclusion): When the ratio of the sum of the cross-sectional area of the branch pipe and the cross-sectional area of the header pipe was reduced by about 30 times, it was confirmed that the flow rate flowing from each branch pipe to the membrane module was almost equal. Also, If the flow in the header pipe is transitional or laminar (Reynolds No. is approximately 4,000 or less), the flowrate flowing from each branch pipe to the membrane module can be more even.

Comment 3:

 (3) Clarify and discuss the motivation of this study, and the novelty and the significance of the results obtained here and compare them with those available in the literature, also including discussions on potential applications.

Reply:

- In this study, the authors pointed out the flow unevenness problem of the currently introduced low-pressure membrane module for drinking water treatment, and focused on the phenomena and causes.

- It was found that flow equalization methods in other fields are not effective for membrane modules introduced to water treatment.

- As the reviewer mentioned, it is too early to propose a new innovative method based on the novelty of this study.

Comment 4:

 (4) Some information is missing in the references section, and hence authors should need to modify it

Reply:

- The authors revised the reference section.

Comment 5:

 (5) What software is used for the simulations? Was the code for the method implemented by the authors or a function already existing in the software was used? If the code for the numerical method was taken from another publication or is part of the software used, please cite the resource.

Reply:

- In this study, the commercial CFD software, ANSYS CFX 16.0 was used.  

- Also, the authors added reference ‘CFX Guideline mannuals’.

Comment 6:

 (6) You should add appropriate references for governing equation.

Reply:

- As your comments, the authors added references.

Comment 7:

 (7) Need more discussion about the results? In the results section, the author needs to analyze the finding by giving reasons for each fact. Please explain every point?

Reply:

- Sorry for the lack of sufficient explanation.

- Considering the current research results, I think that this paper provides information of interest to readers just by identifying that the flow unevenness between parallel low-pressure membrane modules and finding out the cause.

Reviewer 2 Report

I think that the article corresponds to the subject of the journal Membranes. I recommend accepting an article for publication in a journal after following revisions:

  1. It is unclear from the background whether research has been carried out on similar topics before you. In your work, you write "although the studies on the flow distribution of header pipes with manifolds have significantly progressed since the late 1990s, most of them are achievements in fuel cell and energy engineering fields". For CFD, it does not matter where the devices are used: on membrane module piping in the water treatment field or energy engineering fields. Perhaps I overlooked some fundamental differences.
  2. In your work, you write "standard k–ε model was usesd". Why? Are there any works that say that this particular model works well for your tasks? Where is the verification of the adequacy of the results obtained? How was the error in the calculation of results evaluated?  The results obtained in the work are not subject to assessment, since there is no comparison with the classical results, showing the adequacy of the studies carried out.
  3. It is not clear from the manuscript how the dimensions were chosen during the modeling. Are you testing existing equipment or are you planning to design a new one?
  4. The design of the figures needs to be improved. For example, most figures has a lot of empty spaces. Figure 6 is difficult to understand, since at some points there is a large error. For example, Figure 11 b is not understood why the pressure is so the same at all points. It is also incomprehensible why the names on the axes of the figures are written in full, aren't there any generally accepted values?
  5. In the introduction, you write "Finally, the relationship between the Reynolds number in the header pipe and the degree of evenness of the manifold flow distribution was investigated." In the conclusions, you write "Finally, the relationship between the Reynolds number in the header pipe and the degree of evenness of the manifold flow distribution was investigated." and "Therefore, considering the Reynolds number in the header pipe, the flow distribution into the membrane module can be equalized when the flow in the pipe is transitional and laminar (Reynolds number is approximately 4,000 or less)." But in the whole article there is neither a graph nor a figure, where we could see the relationship of the Reynolds number with any other value.

Author Response

Please see the attached file "Reply_for_Reviewer_2_Comments.docx"

Round 2

Reviewer 1 Report

The authors have largely addressed the detailed questions I raised in my previous review. So paper can be accepted in the present form.

Author Response

I appreciate Reviewer 1 for careful review. 

Reviewer 2 Report

The authors reacted very quickly to most of the comments, omitting or ignoring some of them:
1. I still believe that it is impossible to use the turbulence model without checking its adequacy "because everyone uses it". Each scientist has his own problem that needs to be solved, and all the tools used to solve it must have an adequate solution and a minimum error. By the way, I did not see the answer to the question about the error of the results obtained, if there is no estimate of the error of the results, then it is impossible to say whether they are correct. You can also estimate the error of the results obtained by indirect methods.
2. I did not see the answer, why the axes in the figures are indicated not by generally accepted values, but written in full. A nomenclature of variables should be included in the manuscript.
3. Why didn't correct the remark regarding the Reynolds number? I still want to see in manuscript "the relationship between the Reynolds number in the header pipe and the degree of uniformity in the header flow".
